# First-Passage-Time Analysis of the Pulse-Timing Statistics in a Two-Section Semiconductor Laser under Excitable and Noisy Conditions

**Daan Lenstra** * , **Lukas Puts** and **Weiming Yao**

Institute of Photonics Integration, Eindhoven University of Technology, P.O. Box 513,
5600 MB Eindhoven, The Netherlands
* Correspondence: dlenstra@tue.nl; Tel.: +31-6488-75241

**Abstract:** A two-section semiconductor laser can exhibit excitability for certain parameter settings. When used as a photonic spiking neuron, it is relevant to investigate its sensitivity to noise due to, e.g., spontaneous emission. Under excitable conditions, the system emits irregularly timed noise-triggered pulses. Their statistics is analyzed in terms of a first-passage time distribution for the fluctuating intensity to reach the threshold for excitable response. Two analytic approximations valid for short and long times, respectively, are derived which very well explain measured and simulated pulse-repetition time distributions. This provides physical insight into the noise-triggered spiking mechanism.

**Keywords:** semiconductor laser; excitability; first-passage time; pulse timing statistics; optical neuron

## 1. Introduction

A vast spectrum of scientific and engineering fields employ and benefit from increasingly more complex artificial intelligence (AI) algorithms, which in turn drive the demand for faster and more energy-efficient computational hardware. Brain-inspired, neuro-morphic, hardware architectures are being investigated [1,2], among which photonic implementations are studied for their ultra-fast and parallel processing capabilities [3]. Spiking neural network (SNN) hardware has attracted a lot of interest for its similarity with information processing (neural behavior) in the human brain, which is the most energy efficient neural network known and where data are encoded using short electrical signals or spikes [4]. It has been shown that photonic implementations can result in spiking neurons employing optical pulses at a much shorter time scale [5,6]. The spikes in the human brain are digital in amplitude but analogue in time, generated by individual neurons and transmitted through axons [7]. In [4], an extensive comparison between the human brain and digital computing in terms of energy efficiency is given.

The principles of a biological SNN can be transferred to integrated photonics due to the spiking capabilities of semiconductor lasers [5,8,9]. Integrated photonic SNNs benefit from high switching speed, high communication bandwidth, low crosstalk [5,9] and temporal characteristics governed by ultra-fast carrier dynamics [10,11]. This results in the operation of an optical neuron orders of magnitude faster than its biological counterpart. These semiconductor lasers can be the building block in an all-optical SNN on a photonic integrated circuit (PIC).

One successful realization of a photonic spiking neuron is by using an integrated two-section Fabry–Pérot-type (FP) semiconductor laser, where one section operates as the gain and the other as saturable absorber. This configuration is known to exhibit, apart from self-pulsations and CW operation, a form of excitability when operating near, but below threshold [12–14]. In excitable conditions, the laser emits a short optical pulse when triggered by an optical input pulse of sufficiently large energy and thus can operate as an

artificial neuron. We will demonstrate, both experimentally [15] and in simulations, that near the lasing threshold, such an optical trigger can also be caused by a sufficiently large positive intensity fluctuation due to spontaneous-emission noise. As this "spontaneous" emission of output pulses is a source of errors in the functionality of this artificial neuron, it is relevant to investigate this phenomenon in more detail as to its statistics, which is the purpose of the present study.

A semiconductor laser with saturable absorber is accurately described by the well-known Yamada model [12,16,17], which will be used in this paper to investigate the effects of optical noise on the laser's self-spiking behaviour and in simulations. By interpreting the noise-triggered self-pulsations in the excitability parameter region, in terms of first-passage events for the intensity, we derive analytic approximations for the self-pulsation timing statistics, which can be compared with simulated and observed timing statistics [15].

## 2. Device under Study and Observations

The gain and saturable absorber integrated laser was fabricated in a commercially available active–passive multi-project wafer (MPW) InP integration platform [18]. The device basically consists of two electrically isolated semiconductor optical amplifiers (SOAs), one for the gain and the other for the saturable absorption, between two mirrors. The output of the laser is coupled to the edge of the chip, where the emitted light is collected using lensed fibers. A schematic of the two-section laser and a photograph of one of the fabricated devices can be seen in Figure 1. The measurement set up and method are described in [15].

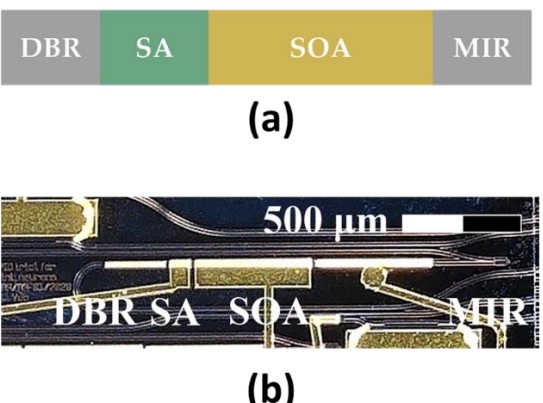

**(a)**

**(b)**

**Figure 1.** (**a**) Schematic overview of a laser with saturable absorber and gain section laser. The absorber and gain elements are surrounded by mirrors (DBR and MIR, see text) to create an optical cavity. (**b**) Micrograph of prototype gain and saturable-absorber laser.

The linear cavity contains an 80 μm saturable absorber, 500 μm gain section and a 500 μm phase shifter. One mirror is a 350 μm distributed Bragg reflector (DBR), optimized for reflections at 1550 nm, and the other a multimode interference reflector (MIR), with estimated reflectivities of 0.76 and 0.40, respectively. The electrical isolation sections are 30 μm in length. In the measurement setup, an optical isolator prevents back reflections from the measurement equipment to the chip (see [15]).

The gain and absorber currents are set to values for which simulations indicate the existence of the excitability regime (see [15]). This corresponds to laser operation just below threshold, which for our two-section laser is 50.54 mA at the saturable-absorber voltage of 0.72 V. To investigate the emission of output pulses in the excitability regime under the influence of spontaneous-emission noise, the absorption is decreased, so that the laser operates even closer to, but still below, threshold and the relative intensity fluctuations become large enough to have a measurable probability to overcome the excitability threshold. The noise-triggered pulses are observed first when the gain section is biased at 50.11 mA and the absorber voltage at 0.720 V. For this setting, the laser starts to generate pulses, without any optical injection. By increasing the absorber voltage slightly, the absorption decreases

and the pulse density increases. In Figure 2, two examples of measured time traces of the laser output at absorber voltages of 0.727 V and 0.730 V are given. These pulses do not occur at a fixed repetition rate, as would be expected for a self-pulsating laser, but occur in a random manner and rather are triggered by spontaneous-emission noise.

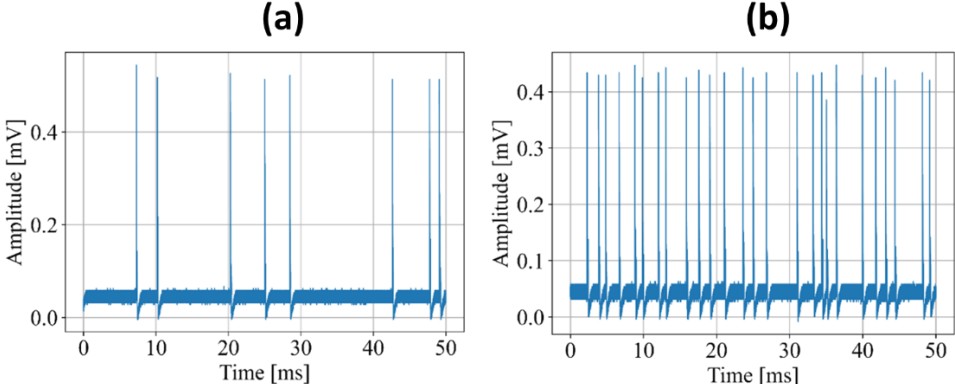

**Figure 2.** Spontaneous-emission-triggered excitable pulses for absorber voltages in (**a**) 0.727 V and (**b**) 0.730 V. In both cases, the gain section is biased at 50.11 mA. From [15].

Figure 3 shows a histogram of observed consecutive pulse timings for an injection current of 50.12 mA for the gain section and voltage 1.38 V for the absorber. The time unit is $\Delta T \equiv 1/S_R$ with $S_R$ the sampling rate of the oscilloscope. The pulse-repetition time distribution is characterized by an initial time interval without pulses, a very steep initial flank and a maximum, followed by a slower decaying tail. We interpret the time at which the pulse-timing distribution starts as the refractory time $T_{refr}$ associated with the excitability at hand [12]. The skew distribution will be analyzed and explained in terms of a first-passage-time distribution in Section 3. Simulations based on the Yamada model (see next section) give rise to pulse timing histograms, that will be shown in Section 4 (see also Figure 13 in [15]), which are similar to the histogram in Figure 3. In the next section, we will derive respective asymptotic expressions for short-time and long-time pulse-timing distributions in terms of first-passage times and explain the typical form of the pulse-timing distribution.

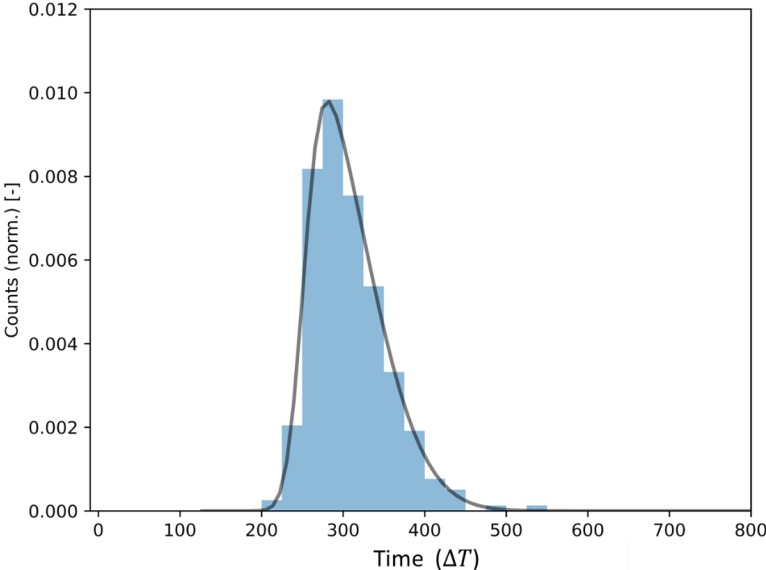

**Figure 3.** Histogram of 313 measured consecutive pulse-timing events. The black line is a calculated fit (skewnorm.fit() function in Python). The refractory time $T_{refr} \approx 200\Delta T$.

### 3. Theoretical Description

The theoretical analysis and simulations are based on the Yamada model [12,16,17],

$$\frac{d}{ds}G = \gamma_G(P_G - G - G\,I); \tag{1}$$

$$\frac{d}{ds}Q = \gamma_A(B_A - Q - \sigma_A Q\,I); \tag{2}$$

$$\frac{d}{ds}I = (G - Q - 1)I + R_s + F_I(s), \tag{3}$$

where all variables and parameters are dimensionless and their meanings summarized in Table 1. The fluctuating Langevin noise term $F_I(s)$ describes the effect of spontaneous emission events on the intensity dynamics and it has the correlation properties [19]

$$< F_I(s) > = 0; \; < F_I(s)F_I(s') > = 2R_s I \delta(s - s'), \tag{4}$$

where $< \ldots >$ denotes the mean value. In fact, $I$, $G$ and $Q$ are stochastic variables and their values given by probability distributions. Denoting the probability distribution for $I$ at time $t$ by $\mathcal{P}(I, t)$, it can be shown that $\mathcal{P}$ is a solution of a Fokker–Planck type diffusion equation [20]

$$\frac{\partial}{\partial s}\mathcal{P}(I, t) = \frac{1}{2}\frac{\partial^2}{\partial I^2}[\mathcal{D}(I)\mathcal{P}(I, t)] - \frac{\partial}{\partial I}[\mathcal{B}(I)\mathcal{P}(I, t)], \tag{5}$$

subject to adequate boundary conditions, and where $\mathcal{D}(I)$ and $\mathcal{B}(I)$ are the diffusion coefficient and drift, respectively, and related to (3) and (4) by

$$\mathcal{D}(I) = 2R_S I, \tag{6}$$

$$\mathcal{B}(I) = (G - Q - 1)I + R_s. \tag{7}$$

**Table 1.** Variables and parameters in the Yamada model.

| Symbol | Name |
|:---:|:---:|
| $s$ | Time (in units $\tau_P$) |
| $\tau_P$ | Cavity photon lifetime |
| $G$ | Gain |
| $Q$ | Absorption |
| $I$ | Intensity |
| $P_G$ | Gain pump parameter |
| $B_A$ | Absorption pump parameter |
| $\gamma_G$ | Gain decay rate |
| $\gamma_A$ | Absorption decay rate |
| $\sigma_A$ | Absorption-to-gain saturation ratio |
| $R_s$ | Spontaneous-emission rate |
| $F_I(s)$ | Langevin noise |

To obtain an explicit expression for $\mathcal{B}(I)$ in terms of $I$, we apply the adiabatic-following solution for $G$ and $Q$, by equating (1) and (2) to 0. This yields $G = \frac{P_G}{1+I}$ and $Q = \frac{B_A}{1+\sigma_A I}$, so that in this adiabatic approximation, we find

$$\mathcal{B}(I) = R_S + \left(\frac{P_G}{1 + I} - \frac{B_A}{1 + \sigma_A I} - 1\right)I. \tag{8}$$

If we then express the stationary solution of (5) as

$$\mathcal{P}_0(I) = \frac{1}{C}e^{-V(I)}, \tag{9}$$

with $C$ a constant normalizing $\mathcal{P}_0$ to unity, that is, $\int_0^\infty dI P_0(I) = 1$ and $V$ the potential that can be associated with the dynamical system (1), (2) and (3) (in the adiabatic approximation), Equation (5), with the right-hand side set to 0, leads to

$$\mathcal{B}(I) = \frac{1}{2}\frac{d}{dI}\mathcal{D}(I) - \frac{1}{2}\mathcal{D}(I)\frac{d}{dI}V(I). \tag{10}$$

Solving (10) for $V(I)$, we find

$$V(I) = \frac{I}{R_S} + \frac{B_A}{R_S\sigma_A}\ln(1 + \sigma_A I) - \frac{P_G}{R_S}\ln(1 + I). \tag{11}$$

where we fixed the integration constant such that $V(0) = 0$. Hence, we can write

$$\mathcal{P}_0(I) = \frac{1}{C}e^{-I/R_S}\frac{(1+I)^{P_G/R_S}}{(1+\sigma_A I)^{B_A/(R_S\sigma_A)}}. \tag{12}$$

In Figure 4, an example is shown for $\mathcal{B}(I)$, $V(I)$ and the corresponding stationary intensity distribution $\mathcal{P}_0(I)$. For this value of the spontaneous-emission rate ($R_S = 10^{-2}$), the intensity is nearly Gaussian distributed with mean $<I> = 1.72$ and FWHM = 0.44.

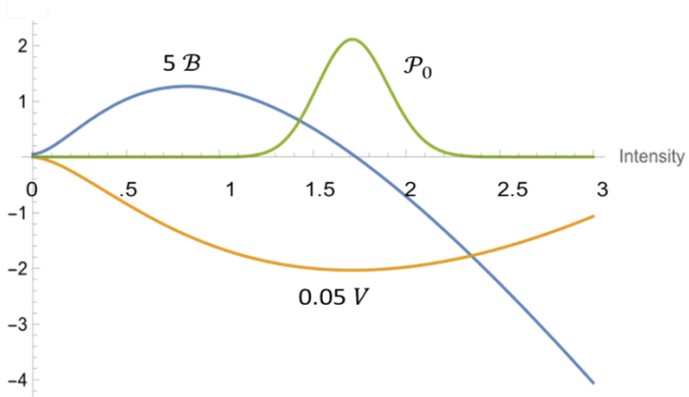

**Figure 4.** Drift $\mathcal{B}$ (blue; $\times 5$), potential $V$ (yellow; $\times 0.05$) and stationary intensity distribution $\mathcal{P}_0$ (green; normalized to one) versus $I$, for the case $R_S = 10^{-2}$; $P_G = 6$; $B_A = 4.95$; $\sigma_A = 1.8$.

Now, we consider the situation where the intensity fluctuations are driven by the diffusion coefficient (6) in the force field described by the drift (7). We are interested in the time $T$ it takes for the intensity, initially at $I_1$, to hit the excitability threshold value $I_C > I_0$ for the first time, where $I_0$ is the most probable intensity, $I_0 \approx <I>$. When this happens, a response pulse will be emitted by the laser. It then takes one refractory period $T_{refr}$ before the laser is ready for the next excitation. This time $T$, the first-passage time (FPT), is obviously a stochastic quantity itself and thus given by a distribution, the first-passage-time density (FPTD) $P_{first}(T; I_1, I_C)$ for the intensity to diffuse from $I_1$ to $I_C$. The pulse timing distribution is then given by $P_{tim}(T; I_1, I_C) = 0$ when $0 < T < T_{refr}$ and $P_{tim}(T; I_1, I_C) = P_{first}(T - T_{refr}; I_1, I_C)$ for $T > T_{refr}$. Strictly speaking, since the refractory time follows a distribution itself, the final pulse repetition statistics is a convolution of the FPTD and the refractory-time distribution. As we have not found an analytical solution for the refractory-time distribution, we will treat $T_{refr}$ as a fixed given quantity.

It is shown in [20] that the FPTD can be expressed entirely in terms of the stationary intensity distribution $\mathcal{P}_0(I)$. The main general results will be repeated here; the full details can be found in [20]. First of all, we introduce the moment-generating function

$$M(z; I_1, I_2) \equiv \int_0^\infty dT e^{zT} P_{first}(T; I_1, I_2), \tag{13}$$

with $z \geq 0$. $M$ is a Laplace transform with negative argument $(s \rightarrow -z)$ and can be derived from a function $G(z; I)$ such that

$$M(z; I_1, I_2) = exp\left\{ \int_{I_1}^{I_2} dIG(z; I) \right\}. \tag{14}$$

Introducing another function

$$K(z; I) \equiv \mathcal{D}(I)\mathcal{P}_0(I)G(z; I), \tag{15}$$

and expressing $K(z; I)$ as a power series in $z$,

$$K(z; I) = \sum_{n=1}^{\infty} K_n(I)z^n, \tag{16}$$

the following result is derived in [20]:

$$K_1(I) = 2 \int_0^I dI' \mathcal{P}_0(I') \tag{17}$$

$$K_n(I) = \int_0^I dI' \frac{\displaystyle\sum_{\substack{p,q \\ p+q=n}} K_p(I')K_q(I')}{\mathcal{D}(I')\mathcal{P}_0(I')}, \qquad (n = 2, 3, 4, \ldots). \tag{18}$$

The problem of determining $M(z; I_1, I_2)$ is, in principle, solved now, apart from the actual calculation of the integrals. It follows directly from definition (13) that the mean first-passage time from $I_1$ to $I_2$ is given by

$$< T >_{I_1,I_2} = \frac{\partial M(z; I_1, I_2)}{\partial z}\Big|_{z=0} = 2 \int_{I_1}^{I_2} \frac{dI}{\mathcal{D}(I)\mathcal{P}_0(I)} \int_0^I dI' \mathcal{P}_0(I'). \tag{19}$$

By repeatedly using (18), the moments can be expressed in multiple integrals, the number of which rapidly increases with increasing order $n$.

So far, (13) to (19) are exact and generally valid. In order to derive an explicit expression for the FPTD, we proceed by making some approximations, based on the observation that the function $\frac{1}{\mathcal{D}(I)\mathcal{P}_0(I)}$ has two sharp maxima on the interval $[0, I_2]$, one at $I = 0$, the other at $I = I_2$, if $I_2 \gg I_0$. Thus, for $I < I_0$, we have

$$K_2(I) = \int_0^I dI' \frac{K_1(I')^2}{\mathcal{D}(I')\mathcal{P}_0(I')} \cong K_1(0)^2 \int_0^I \frac{dI'}{\mathcal{D}(I')\mathcal{P}_0(I')} = 0, \tag{20}$$

implying that $K_n(I) = 0$ for all $n = 2, 3, 4, \ldots$ and

$$M(z; I_1, I_2) \cong exp\left\{ \int_{I_1}^{I_2} dI \frac{K_1(I)\, z}{\mathcal{D}(I)\mathcal{P}_0(I)} \right\} = \exp\{ z < T >_{I_1,I_2} \}, \qquad (I_1 < I_2 \leq I_0), \tag{21}$$

where the last equality follows from (19). An approximated expression for $M$ when $I_C > I_0$ is derived in Appendix A with the result

$$M(z; I_1, I_C) \approx exp\left( < T >_{I_1,I_0} z + \frac{T_{I_0,I_C}\, z}{1 - T_{I_0,I_C}z} \right), \quad (I_C > I_0 > I_1), \tag{22}$$

where $T_{I_0,I_C}$ is a characteristic time related to the first passage from $I_0$ to $I_C$, defined as

$$T_{I_0,I_C} \equiv \frac{K_1(I_C)}{\mathcal{D}(I_C)} \int_{I_0}^{I_C} \frac{dI}{\mathcal{P}_0(I)}. \tag{23}$$

Although $T_{I_0,I_C}$ may be of the same order of magnitude as the average first-passage time (19), it should not be confused with that.

By inverse Laplace transform, (22) with $I_1 = I_0$ leads to the following explicit analytic form for the FPTD:

$$P_{first}(T; I_0, I_C) = e^{-(1 + \frac{T}{T_{I_0,I_C}})} \frac{I_{Bessel,1}\left(2\sqrt{\frac{T}{T_{I_0,I_C}}}\right)}{\sqrt{T_{I_0,I_C} T}}, \quad (T \text{ large}), \tag{24}$$

where $I_{Bessel,1}$ is the modified Bessel function of the first kind. It can easily be checked that (24) is not valid for short times. In fact, for $T \downarrow 0$, (24) yields $P_{first} \to (eT_{I_0,I_C})^{-1}$ (see [21] (p. 375)), whereas on physical grounds, $P_{first}$ should vanish there. Another argument why (24) is an asymptotic expression for large $T$ is given in Appendix A. There, it is argued that the approximation leading to (22) can only be accurate for small $z$, implying large $T$.

For the short-time behavior, we assume that the fluctuating intensity near the mean value $I_0$ behaves under the influence of the spontaneous emission noise as a random walk, with fixed diffusion coefficient $\mathcal{D} = 2R_S I_0$, flat potential $V$ and zero drift, $\mathcal{B} = 0$. For such a case, the probability density for the first-passage time from initial intensity $I_i$ to the absorption point at $I_C > I_i$ in this random-walk approximation is given by [22,23]

$$P_{first}(T; I_i, I_C) = \frac{(I_C - I_i)}{\sqrt{4\pi\mathcal{D}T^3}} e^{-\frac{(I_C - I_i)^2}{4\mathcal{D}T}}, \tag{25}$$

which has a maximum for $T_{max} = \frac{(I_C - I_i)^2}{6\mathcal{D}}$, the most likely first-passage time.

It should be realized that the random-walk approximation for the first-passage time can be correct only for small $T$ but for large $T$, the tail of the distribution will approach zero faster. This is a consequence of the potential not being flat (as in the random walk), but forming a hill slope upward for intensities near the threshold intensity $I_C$. Therefore, the probability for a first passage at larger times $T$ will be smaller than in the random-walk case. Indeed, it can be shown that for large $T$, the tail of the distribution in (24) behaves asymptotically as [21]

$$P_{first}(T; 0, I_C) \underset{T \to \infty}{\to} \frac{e^{-\left(\sqrt{\frac{T}{T_{I_0,I_C}}} - 1\right)^2}}{T_{I_0,I_C}^{1/4} T^{3/4}}. \tag{26}$$

In Figure 5a, the two approximations for $P_{first}(T; I_0, I_C)$ are depicted. The yellow curve is the large-$T$ approximation (24) for $T_{I_0,I_C} = 3.1$ with $I_0 = 1.72$, $I_C = 2.2$ and, $R_S = 0.01$, while the green curve is the random-walk approximation (25). These data pertain to the situation of Figure 4. The two curves connect around $T \sim 6$. This can be seen more convincingly in the logarithmic plots of Figure 5b, which also illustrates that the long-time tail of the distribution decreases faster than exponentially (see (26)), i.e., much faster than the random-walk approximation.

We have applied the asymptotic expressions (24) and (25) to the histogram of measured data (Figure 2), the result of which is depicted in Figure 6. The green curve is the long-time approximation (24), the yellow curve—the short-time approximation (25).

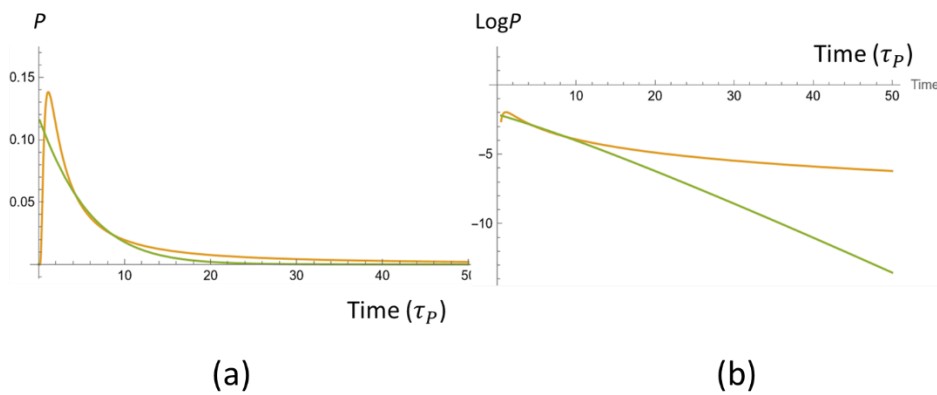

**Figure 5.** Two approximations for $P_{first}(T; I_0, I_C)$ plotted versus time $T$. The green curve is the approximation (24) and is valid for large $T$, while the yellow curve is the random-walk approximation (25), valid for small $T$. In (**a**), the vertical scale is linear; in (**b**), the vertical scale is logarithmic. Parameters are $T_{I_0,I_C} = 3.1$, $I_C = 2.2$, $I_0 = 1.72$, $\mathcal{D} = 2R_S I_0$ and $R_S = 0.01$. These data pertain to the situation of Figure 4. The yellow curve assumes its maximum for $T_{max} = 1.12$.

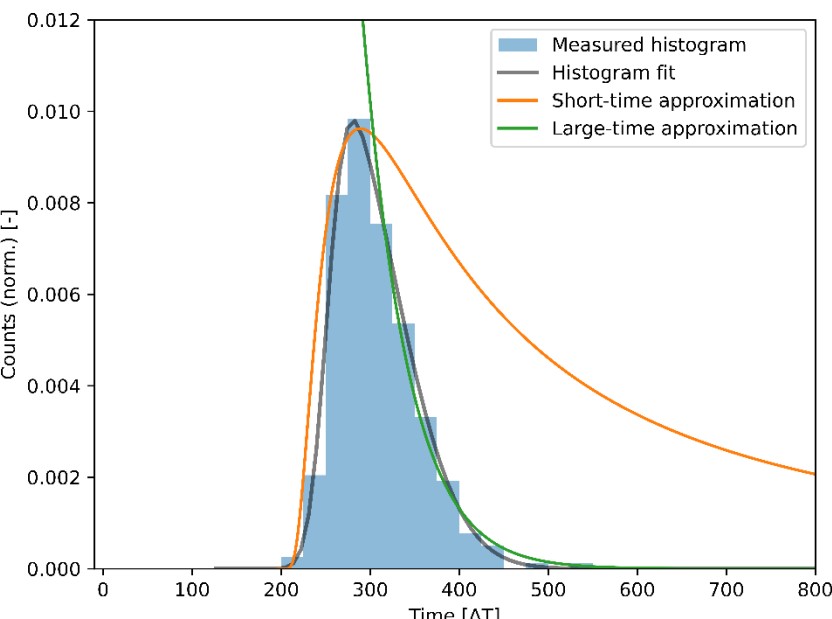

**Figure 6.** The measured histogram of Figure 3 with the large-time approximation (yellow), the short-time approximation (green) and the skewnorm.fit() function Python fit (black). The parameters used for the approximations are $T_{I_0,I_C} = 33.3$, $I_C = 6.1$ and other parameters as in Figure 5.

## 4. Simulations

The results of the simulations are shown in Figure 7 and they demonstrate that in case of sufficiently weak absorption, i.e., $B_A \lessapprox 3.680$, excitable pulses can be triggered by intensity noise due to spontaneous emission in a narrow region very close to and below the laser threshold. The noise results in an irregular train of pulses, of which the density can be controlled by changing the absorption. This can be compared with experimental observations shown in Figure 2. The scenario predicted by theory is in qualitative agreement with the observation, although the horizontal time span in Figure 7, i.e., 20 ns (with $\tau_p = 2 \times 10^{-12}$ s) is more than 6 orders of magnitude smaller than in Figure 2. The explanation for this is as follows: we observe from Figure 7 that an increase of $B_A$ by 0.003 leads to one decade decrease in pulse density. Therefore, it is reasonable to conjecture that the 6 decades of time scale difference can be bridged by upshifting the $B_A$-interval by 0.018. Performing the simulations for the upshifted $B_A$-values would

increase the computation time by at least 6 orders of magnitude and hence be impractical, if not impossible.

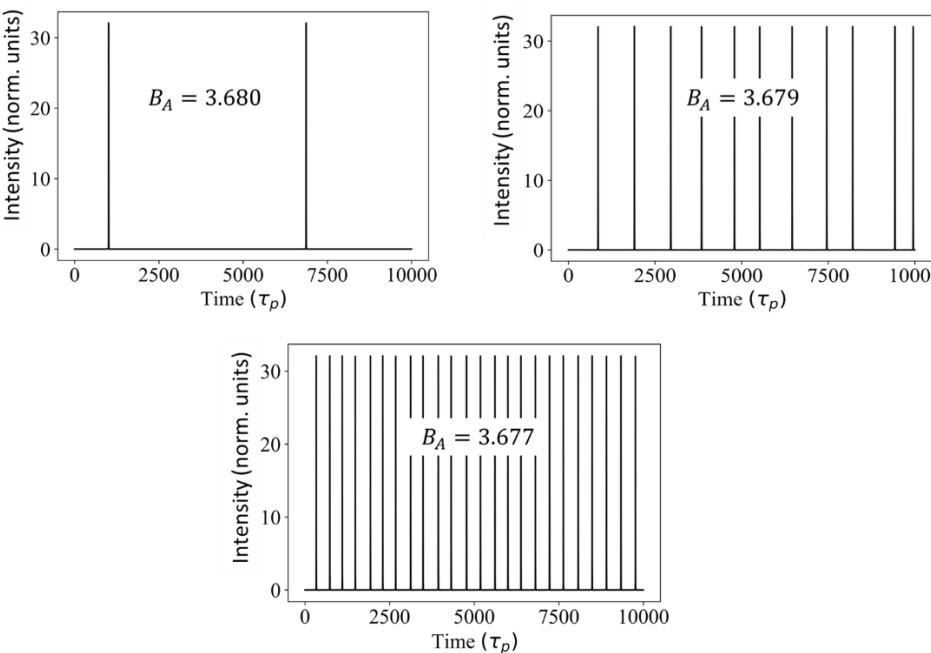

**Figure 7.** Excitable intensity pulses simulated in the Yamada model with noise, i.e., Equations (1)–(3), for fixed gain parameter $P_G = 4.5$ and slightly different values for the absorption $B_A$ as indicated. Other parameters are $R_s = 0.2$, $\gamma_G = 0.05$ and $\gamma_A = 0.1$.

A typical pulse-timing histogram based on a simulated pulse train is shown in Figure 8, for $B_A = 3.6460$, and other parameters as in Figure 7. The black line is the fit provided by Python, the yellow and green curves are the respective approximations (24) and (25) for large and short times.

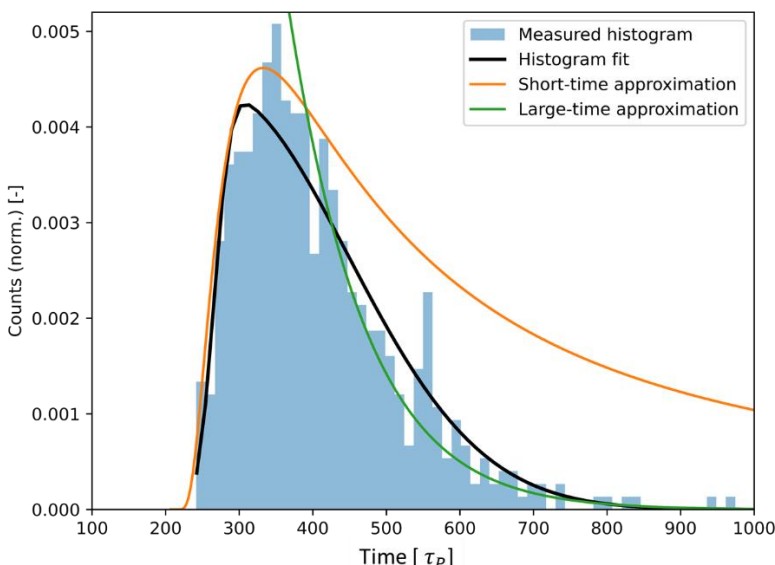

**Figure 8.** Histogram of 583 simulated consecutive pulse-timing events. The black line is a calculated fit (Python). The refractory time $T_{refr} \approx 230\tau_p$. The large-time approximation is the yellow curve, and the short-time approximation—the green curve. The parameters used for the approximations are $T_{I_0, I_C} = 66.7$, $I_0 = 1.0$, $I_C = 18.45$ and $R_S = 0.2$.

## 5. Discussion

We have analyzed and explained the emission of irregularly timed optical pulses from a two-section semiconductor laser with saturable absorber, operating near threshold in a regime of excitability. It is shown that the pulses are triggered by the relatively large spontaneous-emission intensity noise. When such a laser system is used as a neuron, this "spontaneous" emission of output pulses is a source of quasi-random spikes following the derived distribution. It could be employed to simulate random spiking in biological neurons [24], because this phenomenon has similarities to the stochastic behavior of neurons being studied in neuroscience [25–27], with models based on the Fokker–Planck-type equation. This work contributes to the prospect that stochastic neural networks can be simulated with photonics and that problems such as random number generation [28], tunable rate encoding for SNNs [29] or analysis of network population dynamics [30] can be tackled in the future. We have investigated this phenomenon in more detail by focusing on the statistics of the spontaneously emitted pulses.

The numerical simulations and analytical theory are based on the Yamada model for a single-mode semiconductor laser with saturable absorber. By using this model, successful reproduction of observed trends in pulse statistics was obtained in [15]. The observed irregular pulse trains are analyzed in terms of an initial refractory time, whereafter the time interval until the next emitted pulse follows a first-passage-time distribution, for which analytic asymptotic short-time and long-time approximations are derived, which very well explain measured and simulated pulse-repetition time distributions. This provides physical insight into the noise-triggering mechanism.

The noise in the system is mostly dominated by spontaneous emission and its coupling into the lasing mode can be tailored through technology choices, such as the choice of index or gain guiding, variation of the active geometry and the choice of gain material. Hence, the amount of noise can be altered if deterministic spiking is desired.

An interesting research question for further investigation is the observation of coherence resonance [12] in the device studied here. The relevant quantity to investigate then is the normalized jitter $< (T - \langle T \rangle)^2 >^{1/2}/{<T>}$, as a function of the absorber voltage $V_A$, where $<T>$ is the mean pulse-repetition time.

Finally, there are major challenges in moving from a single photonic spiking neuron towards a spiking network on-chip. The output of a single laser neuron needs to be able to trigger another subsequent laser neuron, making it cascadable. This depends on the interconnection architecture, on-chip losses and the required excitation pulse energy. Another challenge is to address on-chip spurious reflections, originating from interfaces and other components in a photonic neural network, prompting for more research in active laser feedback compensation techniques. The third challenge is related to the accurate control of many interconnected laser neurons. This requires high-density electrical interconnects, control electronics and optical monitoring functions to be co-integrated with the photonic chip. Furthermore, low-energy components that perform synaptic weighting on the photonic chip are subject of active research and required in an all-optical SNN.

## 6. Conclusions

Excitable pulse-firing semiconductor lasers with saturable absorber are promising candidates as spiking neurons in an all-optical SNN. Next to desired deterministic spiking when triggered by an input pulse, we observe that these neurons show random output spikes as a consequence of large-enough intensity fluctuation due to spontaneous emission noise. The (average) rate of emission of these noise-induced pulses is investigated in terms of a first-passage time phenomenon. The average time between consecutive pulses is given by $< T >_{I_0, I_C} + T_{refr}$, where $< T >_{I_0, I_C}$ is expressed in (19) in terms of the intensity diffusion coefficient and the stationary intensity probability distribution.

The distribution of spiking events is investigated experimentally and in great detail theoretically. Asymptotic expressions for small and large times are derived and, when confronted with the measured timing statistics, shown to correctly describe the pulse-

timing distribution. The understanding of the influence of spontaneous-emission noise is important for reliable operation of an excitable two-section semiconductor laser when used as a photonic spiking neuron and it can be helpful for studies of stochastic spiking of biological neurons, as well.

**Author Contributions:** Conceptualization, D.L. and W.Y.; methodology, D.L. and W.Y.; software, L.P. and W.Y.; validation, D.L., L.P. and W.Y.; formal analysis, D.L.; investigation, L.P.; resources, W.Y.; data curation, L.P. and W.Y.; writing—original draft preparation, D.L.; writing—review and editing, D.L., L.P. and W.Y.; visualization, D.L. and L.P.; supervision, W.Y.; project administration, W.Y.; funding acquisition, W.Y. All authors have read and agreed to the published version of the manuscript.

**Funding:** This work is supported by the Netherlands Organization for Scientific Research (NWO): Veni grant (17269) 'Light up the brain: accelerating AI research with integrated photonics' and Netherlands Organization for Scientific Research (NWO): Zwaartekracht Grant 'Research Center for Integrated Nanophotonics'.

**Institutional Review Board Statement:** Not applicable.

**Informed Consent Statement:** Not applicable.

**Data Availability Statement:** Measurement results and details on the measurement equipment can be found in [14,15].

**Acknowledgments:** The authors thank Kevin Williams for his stimulating support.

**Conflicts of Interest:** The authors declare no conflict of interest. The funders had no role in the design of the study; in the collection, analyses, or interpretation of data; in the writing of the manuscript, or in the decision to publish the results.

## Appendix A. Derivation of Approximate Expression for $M(z;I_1,I_2)$ When $I_0 \leq I_1 \leq I_2$

We can now approximate $K_2(I)$ (see (18)) for $I > I_0$ by

$$K_2(I) = \int_{I_0}^{I} dI' \frac{K_1(I')^2}{\mathcal{D}(I')\mathcal{P}_0(I')} \cong \frac{K_1(I)^2}{\mathcal{D}(I)} \int_{I_0}^{I} \frac{dI'}{\mathcal{P}_0(I')}. \tag{A1}$$

and $K_3(I)$ by

$$
\begin{aligned}
K_3(I) \quad &= \int_0^I dI' \frac{2K_1(I')K_2(I')}{\mathcal{D}(I')\mathcal{P}_0(I')} = 2\left(\int_0^{I_0} dI' + \int_{I_0}^I dI'\right) \frac{K_1(I')K_2(I')}{\mathcal{D}(I')\mathcal{P}_0(I')} \\
&\cong 2\int_{I_0}^I dI' \frac{K_1(I')}{\mathcal{D}(I')\mathcal{P}_0(I')} \left(\int_0^{I_0} dI'' + \int_{I_0}^{I'} dI''\right) \frac{K_1(I'')^2}{\mathcal{D}(I'')\mathcal{P}_0(I'')} \\
&\cong 2\int_{I_0}^I dI' \frac{K_1(I')}{\mathcal{D}(I')\mathcal{P}_0(I')} \int_{I_0}^{I'} dI'' \frac{K_1(I'')^2}{\mathcal{D}(I'')\mathcal{P}_0(I'')} \\
&\cong 2\frac{K_1(I)^3}{\mathcal{D}(I)^2} \int_{I_0}^I \frac{dI'}{\mathcal{P}_0(I')} \int_{I_0}^{I'} \frac{dI''}{\mathcal{P}_0(I'')} = \frac{K_1(I)^3}{\mathcal{D}(I)^2} \left(\int_{I_0}^I \frac{dI'}{\mathcal{P}_0(I')}\right)^2, \quad (I > I_0).
\end{aligned} \tag{A2}
$$

In the derivation of (A2), we use several times the approximation mentioned just above (20). Similarly, for $n = 3, 4, 5, \ldots$ we obtain

$$K_n(I) \approx \frac{K_1(I)^n}{\mathcal{D}(I)^{n-1}} \left(\int_{I_0}^{I} \frac{dI'}{\mathcal{P}_0(I')}\right)^{n-1}. \tag{A3}$$

Hence we derive, using (16),

$$K(z;I) \approx K_1(I)z \sum_{n=0}^{\infty} \left(\frac{K_1(I)z}{\mathcal{D}(I)} \int_{I_0}^{I} \frac{dI'}{\mathcal{P}_0(I')}\right)^n = \frac{K_1(I)z}{1 - \frac{K_1(I)z}{\mathcal{D}(I)} \int_{I_0}^{I} \frac{dI'}{\mathcal{P}_0(I')}}, \quad (I > I_0). \tag{A4}$$

Since the cumulative error in the approximation (A3) increases with increasing order $n$, (A4) should not be taken too seriously for large $z$. Defining a characteristic time $T_{I_1,I_2}$ related to the first passage from $I_1$ to $I_2$, i.e.,

$$T_{I_1,I_2} \approx \frac{K_1(I_2)}{\mathcal{D}(I_2)} \int_{I_1}^{I_2} \frac{dI}{\mathcal{P}_0(I)}, \tag{A5}$$

we can express $K(z;I)$ as

$$K(z;I) \approx \frac{K_1(I)z}{1 - T_{I_0,I}z}, \tag{A6}$$

from which we obtain

$$G(z;I) \approx \frac{K_1(I)z}{\mathcal{D}(I)\mathcal{P}_0(I)} \frac{1}{1 - T_{I_0,I}z}, \tag{A7}$$

and

$$\begin{aligned} M(z;I_1,I_2) &\approx exp\left( \int_{I_1}^{I_2} dI \frac{K_1(I)z}{\mathcal{D}(I)\mathcal{P}_0(I)} \frac{1}{1 - T_{I_0,I}z} \right) \\ &\approx exp\left( \frac{T_{I_1,I_2}\,z}{1 - T_{I_0,I_2}z} \right), \quad (I_2 \geq I_1 \geq I_0). \end{aligned} \tag{A8}$$

Finally, using (see (14))

$$M(z;I_1,I_C) = M(z;I_1,I_0)M(z;I_0,I_C), \tag{A9}$$

we find with (21) and (A8)

$$M(z;I_1,I_C) \approx exp\left( <T>_{I_1,I_0} z + \frac{T_{I_0,I_C}\,z}{1 - T_{I_0,I_C}z} \right), \ (I_C \geq I_0 > I_1), \tag{A10}$$

which is the desired expression given in (22). Note that $T_{I_0,I_C}$ should not be identified with $\langle T \rangle_{I_0,I_C}$.

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
