# Peer review of "First-Passage-Time Analysis of the Pulse-Timing Statistics in a Two-Section Semiconductor Laser under Excitable and Noisy Conditions"

_photonics, doi:10.3390/photonics9110860_

Round 1

Reviewer 1 Report

In their manuscript, Lenstra et al. study the excitability of a two-section semiconductor laser to the aim of interpreting the system as a photonic spiking neuron, in analogy to the cells of brain-inspired neural networks.

The device under study is an integrated two-section Fabry-Perot semiconductor laser with one section operating as the gain and another section operating as the absorber. The study focuses the most on the statistics of the times at which excitable pulses occur as an effect of spontaneous emission noise near lasing threshold, both in terms of first-passage statistics and in terms of pulse-timing. As a result of some approximations the authors are able to find an analytical expression for both aspects, starting from a simple stochastic Yamada model. The analytical result well describes experimental findings and the result of numerical simulations. While the occurrence of these excitable pulses due to spontaneous emission noise could in principle be the cause of errors in artificial neural networks, the authors also argue that this kind of stochatistic behavior is more properly descriptive of the stochastic spiking behavior exhibited by biological neurons.

The manuscript is an interesting addition to the current state of the art in the interpretation of laser dynamics in the framework of excitable spiking neuron behavior. 

I would recommend publication with minor corrections provided that the authors properly answer the following questions:

1) in terms of novelty: I've noticed that the authors keep referencing paper [17] for most of the results. However paper [17] is currently under review and not accessible. Could the author comment on the differences between the current manuscript and paper [17]?

2) in relation to the first point while I understand citing [17] for the results that have already been included in that paper I would recommend at least including a figure of the fabricated device in the current manuscript, especially because the other paper is not accessible and including this additional figure would make the reading of the current manuscript more self-consistent.

3) I suggest in the second page to indicate the lasing threshold current bias value for the experimental results in Fig. 1.

4) Could the authors comment on how does the distribution in Fig. 2 relate to a Poisson distribution?

5) The parameter tau_p is indicated at line 257 and in Fig. 6 but I don't think it has been introduced. I also suggest to the authors to arrange the scale of the x-axis in Fig. 6 so that it is in the same units as Fig. 1 and not in units of tau_p to improve visual comparison. Alternatively I suggest writing explicitely in the text or caption that the x-axis is in units of tau_p. Could the authors also comment on the reason why this simulation has not been performed for the upshifted value of B_A as they mention?

6) Regarding Fig. 7 I suggest writing explicitely the units of the x-axis. While the result of Fig. 7 is already acceptable and indicative of the validity of the theoretical analysis, I personally would have expected an even better agreement between the histogram and the short-time/long-time approximation curves. Could the authors comment on this point? What is the total number of events that has been considered for Fig. 7?

7) I find particularly interesting the point raised in the discussion section regarding the potential for this kind of laser to better simulate the behavior of stochastic spiking biological neurons. On the other hand both in the introduction and in the conclusions of the manuscript the author discuss the scope of the study mainly as the necessary investigation of a potential problem for the development of an all-optical SNN, since spontaneous noise may lead to erroneous emission. How would this problem be treated when trying to develop a network that mimics in a more correct way the stochastic behavior of spiking biological neurons? 

On the other hand, do the authors have any suggestion on how the "erroneous excitation" could be avoided?

8) Finally, the manuscript focuses on the behavior of a single cell as a spiking neuron. Could the authors comment on what would be the main challenges in the development of a full network using this kind of devices?

Further minor points.

While reading I've noticed the following:

a) the acronym SNN (which I assume stays for Spiking Neural Network) is used in the first page but not introduced.

b) In the introduction of the moment-generating function the variable z is used but not properly introduced. 

c) I assume there is a typo at line 205 for the exponential expression

d) lines 242-243 describing Fig. 5 are not consistent with Fig. 5 and its caption: I imagine the "light-blue" curve to be the "yellow"/orange curve identifying the long-term approximation and the "yellow"/orange curve to be instead the green curve identifying the short-term approximation.

In conclusion, once the previous points have been properly addressed I believe the manuscript to be suitable for publication on Photonics.

Author Response

see the uploaded file

Reviewer 2 Report

In this paper, it is numerically shown that a spontaneous emission leads to the statistic distribution of pulse-timing in a self-pulsating semiconductor laser. I think that the paper contains some new materials worth for publishing in ``Photonics’’ journal. However, I made some comments on the manuscripts.

1. In line 355, what does SNN stand for? Spiking neural network?

2. In line 44, the order of the references is incorrect. The authors should check numbering of the references.

3. In line 71, the authors forget the indentation.

4. Considering the readers, it would be nice to have a diagram showing the structure of the two-section laser in section 2.

5. The reference numbers in the caption of Figure 1 appear to be incorrect.

6. In line 99, the authors describe ``Fig.6’’. But it may be an error of ``Fig. 7’’.

7. In the caption of Fig. 2 and other, the authors describe the histograms have been fitted using Python. I think the algorithm used is more important than the programming language.

8. In line 137, the authors describe ``with C a constant normalizing P_0 to unity’’. In the caption of Fig.3, ``P_0 (green; normalized to one)’’. However, it seems that P_0 in Fig. 3 has the peak 2.0. Which is correct?

9. In the caption of Fig. 3, I think the authors should fix the notation of the exponent.

10. In Fig. 3, 5 and 7, there appear to be a white box or text boxes. This may create suspicion that they were added to modify the diagram. The authors should remake the figures.

11. In Fig. 2, 4, 5 and 7, what does the horizontal axis show? What is the unit of measure? Or are they normalized time S?

12. I do not understand the basis for what you are stating in lines 257-261. Is it shown by anything to change to linear? 

Author Response

see the uploaded file

Reviewer 3 Report

The authors perform a detailed study of excitable pulses emitted by a two-section semiconductor laser, based on the well-known Yamada model. Specifically, the authors study the effect of spontaneous emission noise in the statistics of the timing of excitable optical pulses using first-passage-time analysis. The manuscript is clearly written and the topic is relevant and timely due to the potential application of the excitable laser as a photonic neuron. I am happy to recommend the acceptance of this manuscript, provided that the authors take into account the following points

11)     In the abstract the authors should include at least one sentence that summarizes the main results/findings of the study.

22)     In Fig. 2, the units of the horizontal axis should be consistent with those in Fig. 1: Time interval between consecutive pulses, in ms. In the Fig. caption, the refractory time should also be indicated in ms.

33)     For the sake of clarity, could the authors include a figure that shows the simulated optical pulses (as Fig. 1)?

44)     The label of the horizontal axis in Fig. 5 is missing; the size of the letters in the vertical and horizontal axis should be the same.

55)     In Fig. 6, the authors should indicate what is plotted in the vertical axis (Amplitude in normalized units?)

66)     In Fig. 7, the size of the letters in the vertical and horizontal axis should be the same.

77)     Can the authors discuss about the possibility of observing “coherence resonance” (Ref. 12)? Can this be an interesting research question for future work?

Author Response

see the uploaded file
